# Synthesis and Biological Activities of Luminescent 5,6-Membered Bis(Metallacyclic) Platinum(II) Complexes

**DOI:** 10.3390/molecules28176369

**Published:** 2023-08-31

**Authors:** Jing Jing, Miao Yu, Lei Pan, Yang Zhao, Guo Xu, Hua-Hong Zhang, Chen Li, Xiao-Peng Zhang

**Affiliations:** Key Laboratory of Water Pollution Treatment & Resource Reuse of Hainan Province, College of Chemistry and Chemical Engineering, Hainan Normal University, Haikou 571158, China; 20212070300017@hainnu.edu.cn (J.J.); yumiaonpc@126.com (M.Y.); 202107070731@hainnu.edu.cn (L.P.); 20202070300133@hainnu.edu.cn (Y.Z.); 202212070300022@hainnu.edu.cn (G.X.); 202211070300008@hainnu.edu.cn (H.-H.Z.); 202312070300022@hainnu.edu.cn (C.L.)

**Keywords:** platinum(II) complexes, 5,6-membered bis(metallacycle), biological activities, emission, crystal structures

## Abstract

Four couples of 5,6-membered bis(metallacyclic) Pt(II) complexes with acetylide and isocyanide auxiliary ligands have been prepared and characterized. The structures of (−)-**2** and (−)-**3** are confirmed by single-crystal X-ray diffraction, showing a distorted square-planar coordination environment around the Pt(II) nucleus. Both solutions and solid samples of all complexes are emissive at RT. Acetylide-coordinated Pt(II) complexes have a lower energy emission than those isocyanide-coordinated ones. The emission spectra of N^N′*C-coordinated Pt(II) derivatives show a lower energy emission maximum relative to N^C*N′-coordinated complexes with the same auxiliary ligand. Moreover, the difference between cyclometalated N^N′*C and N^C*N′ ligands exerts a more remarkable effect on the emission than the auxiliary ligands acetylide and isocyanide. Cytotoxicity and cell imaging of luminescent 5,6-membered bis(metallacyclic) Pt(II) complexes have been evaluated.

## 1. Introduction

Cyclometalated N^C^N′- and N^N′^C-coordinated Pt(II) complexes have received significant attention recently due to their great potential in chemical, biological, and optoelectronic applications [1,2,3,4,5,6]. When the classical five-membered metallacycle (C^N) is extended to a six-membered chelating metallacycle (C*N) using an amine connector, N^C*N′- and N^N′*C-coordinated Pt(II) complexes show a much more perfect square-planar geometry compared to N^C^N′- and N^N′^C-coordinated Pt(II) complexes [7,8]. Through tuning coordination geometry, the rectified square structure of a 5,6-membered bis(metallacycle) may maximize the strength of the ligand field and minimize the nonradiative metal-centered transitions, so N^C*N′-, N*N′^C-, and C^N*N′-coordinated complexes are highly emissive with an appreciable emission quantum efficiency (Φ_em_) [9,10]. Strassert et al. investigated oxygen-sensing behaviors of porous materials that phosphorescent (C^N*N′)Pt(II) complexes encapsulated in Zn-based metal–organic frameworks [11]. In addition, Huo et al. demonstrated that a 5,6-membered bis(metallacycle) of platinum complexes could be selectively formed through either the sp^2^ or the sp^3^ C–H bond activation, which can be used in selective C−H bond activation, C−N bond cleavage, and selective acylation [12,13,14].

Although the square geometry is improved by forming a 5,6-membered bis(metallacycle), introducing a 5,6-membered bis(metallacycle) would increase molecular flexibility, which is adverse to the emission efficiency. The rectified geometry may alone seem not to result in a large d orbital splitting enough to eliminate thermally accessible nonradiative d–d transition, and N^C*N′- and N^N′*C-coordinated Pt(II) chloride complexes may be weakly emissive [7,8]. Acetylide and isocyanide ligands induce a much stronger ligand field than the chloride, which could efficiently raise the nonradiative d−d transition to higher energy, leading to a big jump in emission efficiency in cyclometalated N^C^N′- and N^N′^C-coordinated Pt(II) complexes [15,16]. Therefore, introducing strong donors such as acetylide and isocyanide may be a suitable alternative. The emissions of N^C*N′- and N^N′*C-coordinated Pt(II) phenylacetylene and isocyanide derivatives are highly intensive, with Φ_em_ exceeding 40% [9]. The phosphorescent phenylacetylene derivatives could be utilized as triplet photosensitizers for triplet-triplet annihilation upconversion [10].

In previous work, we reported a series of pinene-fused N^C^N′-, N^N′^C-, N^C*N′-, and N^N′*C-coordinated Pt(II) complexes and investigated their solvent-, mechano-, vapor-, and thermo-induced color, luminescent, and chiroptical switching behaviors [17,18,19]. Pinene-containing N^C*N′- and N^N′*C-coordinated Pt(II) complexes displayed significant AIE properties in the THF-water solution system by restricting intramolecular vibration and rotation in the poor solvent. A reversible mechanochromic luminescence was found for the N^N′*C-coordinated Pt(II) chloride complex, and its potential utility in an anti-counterfeiting application was attempted [19]. As an extension, acetylide and isocyanide ligands are introduced in pinene-containing N^C*N′- and N^N′*C-coordinated Pt(II) complexes (Figure 1). Four pairs of 5,6-membered bis(metallacyclic) Pt(II) complexes were characterized by NMR, HRMS, and elemental analysis, and the structures of (−)-**2** and (−)-**3** were unambiguously confirmed by single-crystal X-ray diffraction. We studied all obtained complexes’ photophysical and luminescent properties and explored their cytotoxic properties and applications in cell imaging.

## 2. Results and Discussion

### 2.1. Synthesis and Characterization

The pinene-containing N^C*N′- and N^N′*C-coordinated Pt(II) precursors Pt(N^C*N′)Cl and Pt(N^N′*C)Cl featuring a fused 5,6-membered bis(metallacycle) were prepared according to our reported procedures via Kröhnke strategy and palladium-catalyzed C−N cross-coupling or Cu-catalyzed N-arylation reaction [19]. Cationic Pt(II) isocyanide derivatives (−)-**1** and (−)-**2** were obtained through ligand metathesis reaction between 5,6-membered bis(metallacyclic) chloride precursors and the isocyanide in the presence of AgOTf [18]. Neutral Pt(II) acetylide complexes (−)-**3** and (−)-**4** can be separated by reacting chloride precursors with phenylacetylene [9,20]. NMR (Appendix A), HRMS Appendix A), and EA accurately determined all the target products. Compared to those of the chloride precursors Pt(N^C*N′)Cl and Pt(N^N′*C)Cl, aromatic proton signals of the complexes incorporating isocyanide and acetylide are shifted upfield. Due to *N*-phenyl presence, more aromatic proton signals are observed in 5,6-membered bis(metallacyclic) platinum(II) complexes relative to 5,5-membered counterparts. Characteristic aliphatic proton signals of pinene groups in the high-field regions (0.6–3.5 ppm) are perceived, with –CH_3_ of the pinene skeleton located at ca. 0.70 and 1.45 ppm. Moreover, structures of (−)-**2** and (−)-**3** were confirmed by single-crystal X-ray diffraction. The enantiomers (+)-**1**, (+)-**2**, (+)-**3,** and (+)-**4** were also prepared using the same method.

### 2.2. Crystal Structures

The crystals suitable for X-ray diffraction were separated in mixed CH_2_Cl_2_/acetone (*v*/*v* = 1:1) solutions. The yellow blocks of complex (−)-**2** crystallize in the *P*1 space group of the triclinic system, while the yellow ones of complex (−)-**3** are located in the *P*2_1_ space group of the monoclinic system (Table 1). Both yellow crystals emit orange-yellow luminescence under UV radiation (*λ* = 365 nm). As shown in Figure 1, two molecules involved in the unit cell of both (−)-**2** and (−)-**3** present as an antiparallel-aligned dimer through weak *π*–*π* contacts (3.9 Å) between electron-deficient pyridine ring and electron-rich phenyl moiety using a head-to-tail fashion.

As listed in Appendix A, the bond distances and bond angles between N^C*N′- and N^N′*C-coordinated Pt(II) complexes show a distinct difference. The Pt-C(aryl) distances (2.050(8) and 2.052(8) Å) and Pt-C(isocyanide) distances (1.89(2) and 1.97(2) Å) of (−)-**2** are comparable to those in the reported Pt(N^N′^C)C≡N derivatives [21,22,23]. Due to chelating ring strain, Pt-C(aryl) bond lengths are close to Pt-N(*trans* position) distances. For complex (−)-**3**, two carbon atoms are located at a *trans* position, and Pt-C(aryl) distances (1.994(9) and 1.986(9) Å) differ Pt-C(isocyanide) bonds (2.049(10) and 2.053(9) Å) by less than 0.05 Å, owing to a stronger ligand of the aryl carbon atom [9,24]. Both Pt atoms in complexes (−)-**2** and (−)-**3** reside in a distorted square-planar environment. However, the tridentate chelating strain in the flexible 5,6-membered bis(metallacycle) is relieved relative to the five-five-membered ring, correspondingly the degree of distortion in N^C*N′- and N^N′*C-coordinated Pt(II) complexes is alleviated [7,8,9,19]. In addition, the C1-Pt1-N1 and C3-Pt2-N3 chelating angles (171.9(6) and 172.7(6)°) in N^N′*C-coordinated (−)-**2** are close to more linearity compared with those angles (168.0(4) and 168.7(4)°) in N^C*N′-coordinated (−)-**3** (Appendix A).

The isocyanide aryl ring is almost coplanar with the Pt(N^N′*C) moiety in (−)-**2** with dihedral angles being 0.948 and 2.643° (Appendix A), which resembles those in cyclometalated Pt(II) isocyanide complexes [21,22,23]. In contrast, phenylacetylene is commonly twisted with Pt(II) coordination plane, and the torsion angles between phenylacetylene and Pt(N^C*N′) unit in (−)-**3** are 36.817 and 41.976° [9]. Moreover, the Pt-C(alkynyl) bond bends distinctly away from the Pt(N^C*N′) plane with a distance of 0.32 Å between the C(alkynyl) atom and the plane, while the C(isocyanide) atom deviates from the Pt(N^N′*C) plane by 0.18 and 0.10 Å (Appendix A). To minimize the steric interaction between the uncyclometalated *N*-phenyl and neighboring cyclometalated aryl groups in flexible 5,6-membered bis(metallacyclic) Pt(II) complexes [7,8,9,19], the *N*-phenyl group is significantly twisted with the Pt(II) coordination plane with torsion angles extending from 73.15° to 79.71° (Appendix A). The amino nitrogen in a flexible six-membered metallacycle is 0.20–0.37 Å from the platinum-centered plane (Appendix A). No substantial intermolecular Pt⋯Pt interactions are detected in either (−)-**2** or (−)-**3**. The dimers in (−)-**2** are aligned alternately into a one-dimensional head-to-tail chain through weak *π*–*π* contacts along the *a*-axis (Figure 2). However, any intermolecular interactions are not found to connect the neighboring dimers in (−)-**3**, and the dimeric pairs are separated (Figure 2).

### 2.3. Spectroscopic Properties

The solution spectroscopic properties of all the complexes are shown in Figure 3. Intense absorptions (*ε* ≈ 10^4^ L·mol^−1^·cm^−1^) below 350 nm are ascribed to intraligand ^1^*π*,*π** transitions, and several moderate absorption bands (*ε* > 10^3^ L·mol^−1^·cm^−1^) in the low-energy region are attributed to mixed charge transfer transitions of ^1^MLCT (metal-to-ligand charge transfer)/^1^ILCT (intraligand charge transfer) [7,8,9,10,19]. In comparison to isocyanide Pt(II) derivatives ((−)-**1** and (−)-**2**), the low energy absorption of acetylide-coordinated Pt(II) complexes ((−)-**3** and (−)-**4**) with the same 5,6-membered bis(metallacycle) tail up to a longer wavelength. For 5,6-membered bis(metallacyclic) Pt(II) complexes with the same auxiliary ligand, N^N′*C-coordinated Pt(II) derivatives ((−)-**2** and (−)-**4**) show a lower energy absorption relative to those N^C*N′-coordinated complexes ((−)-**1** and (−)-**3**). The low energy absorptions of all complexes are concentration-independent and remain unchanged with increasing concentration (Appendix A), suggesting the absence of aggregated states formed through Pt···Pt interactions in concentrated solution. The solvent influence on absorption has also been studied. The low energy absorption bands provide a slightly hypsochromic shift with the increased polarity of the solvents (Appendix A), confirming the charge transfer (CT) character [25,26]. The chiral characteristics of all complexes have been determined through ECD (electronic circular dichroism) spectra (Appendix A). Mirror-symmetric profiles can be detected for enantiomers. Positive Cotton effects below 350 nm that originate from chiral pinene ligands are observed for (−)-**1**, (−)-**2**, (−)-**3,** and (−)-**4**. In addition, weak Cotton effects in the low-energy region are exhibited [19]. Chiral environments around the central Pt nucleus are insignificant due to the nearly square-planar geometry of 5,6-membered bis(metallacyclic) Pt(II) complexes.

The CH_2_Cl_2_ solutions of N^C*N′-coordinated Pt(II) derivatives (−)-**1** and (−)-**3** emit in the green-yellow region at RT, while the solutions of N^N′*C-coordinated Pt(II) complexes (−)-**2** and (−)-**4** emit orange light (Figure 3). Both complexes (−)-**1** and (−)-**3** display a vibronically structured emission with *λ*_max_ at 493 and 511 nm and shoulders at 528 and 541 nm, respectively (Figure 3 and Table 2). In contrast, a nearly structureless band in the orange region with lower energy emission peaks at 578 and 595 nm is perceived for complexes (−)-**2** and (−)-**4**, respectively. Acetylide-coordinated Pt(II) complexes have a lower energy emission in CH_2_Cl_2_ solution than those isocyanide-coordinated ones (Δ*λ* = ca. 20 nm) (Figure 3 and Table 2), which is consistent with their absorption spectra. In addition to this, the emission spectra of N^N′*C-coordinated Pt(II) derivatives show a lower energy emission maximum relative to N^C*N′-coordinated complexes with the same auxiliary ligand (Δ*λ* = ca. 80 nm) (Figure 3 and Table 2). The difference between cyclometalated N^N′*C and N^C*N′ ligands exerts a more remarkable effect on the emission than the auxiliary ligands acetylide and isocyanide. The emission dependences on concentration and solvent are insignificant (Appendix A). When changing from toluene to methanol, the solvatochromic emission shift is almost less than 10 nm, and the emission profiles stay the same with increasing concentration.

All the complexes provide a more structured and narrower emission band in frozen glass at 77 K than in a fluid at RT (Appendix A). The complexes ((−)-**1** and (−)-**3**) with N^C*N′ skeleton show a negligible rigidochromic effect with the value of emission maximum shift 5–8 nm. In contrast, the rigidochromic shift of derivatives ((−)-**2** and (−)-**4**) with N^N′*C parent ligand is evident with a 22–34 nm difference between the emission maximum in the fluid and the rigid glass. According to the reported investigations on 5,6-membered bis(metallacyclic) Pt(II) complexes, not only the cyclometalated parent ligand (N^C*N′ and N^N′*C) and the auxiliary ligand (isocyanide and acetylide) could affect the emission of complexes but also the conjugation degree of the 5,6-membered bis(metallacycle) through the amino N atom exerts a considerable influence on the emission [8,19]. The emissive state of all complexes can be assigned to a ligand-centered (LC) triplet transition (^3^*π*,*π*) with some CT transitions [7,8,9,19]. The low energy emissions (578 and 595 nm) have more CT character than the high-energy ones (493 and 511 nm) [8,19]. For complexes (−)-**2** and (−)-**4**, the geometric change to maximize conjugation of the 5,6-membered bis(metallacycle) through the amino N atom is unfavorable in the rigid matrix. Hence, the emission state has more LC ^3^*π*,*π* characteristic in the rigid glass at 77 K, showing a high energy emission.

The solids of all complexes are emissive from the green-yellow to orange regions at RT (Figure 4 and Table 2). The emission spectra of N^C*N′-coordinated complexes are more structured and show a higher energy emission maximum than the ones of N^N′*C-coordinated derivatives. At 77 K, the emission profiles become highly structured and narrow. The solids show higher emission quantum efficiencies (Φ_em_) than the solutions (Table 2). The N^N′*C-coordinated derivatives with isocyanide and acetylide could not form aggregates under external stress like chloride precursors, and the mechanochromic luminescence phenomenon is not realized for (−)-**2** and (−)-**4 [19]**.

### 2.4. Theoretical Investigation

Time-dependent density functional theory (TD-DFT) calculations have been performed to explore the origin of the transitions [27]. As revealed in Figure 5 and Appendix A, the nature of S_0_→S_1_ transition of all complexes mainly derive from HOMO (the highest occupied molecular orbital)→LUMO (the lowest unoccupied molecular orbital) with the overwhelming contribution of over 90%. Also, the S_0_→S_1_ transition of complex (−)-**1** involves some composition of HOMO-1→LUMO (7.0%). Based on molecular orbital (MO) patterns and orbital composition analysis (Figure 5 and Appendix A), the HOMO of complexes (−)-**1** and (−)-**2** is mainly delocalized on **Ring B**, **Ring C**, and their bridging atom amino N with a total contribution of over 80%. Their LUMO concentrates on phenyl-pyridine or dipyridine moiety (**Ring A** and **Ring B**). In addition, the central Pt atom in (−)-**1** and (−)-**2** contributes little to HOMO and LUMO.

For acetylide-coordinated Pt(II) complexes (−)-**3** and (−)-**4**, the HOMO primarily comes from Pt nucleus and phenylacetylene with a total contribution of over 60%, and the fragments **Ring B** and **Ring C** also contribute some to the HOMO (Figure 5 and Appendix A). Similar to the LUMO distribution of (−)-**1** and (−)-**2**, the phenyl-pyridine or dipyridine moiety (**Ring A** and **Ring B**) gives a dominant contribution (>75%) to the LUMO of (−)-**3** and (−)-**4**. Therefore, a mixture of ^1^ILCT (N^C*N′ or N^N′*C), ^1^MLCT (from Pt(II) atom to N^C*N′ or N^N′*C) and ^1^LLCT (from phenyl isocyanide or phenylacetylene to N^C*N′ or N^N′*C) transitions should be responsible for the lowest energy absorption band in the UV-vis spectra of all complexes. Furthermore, more ^1^MLCT and ^1^LLCT components are involved in acetylide-coordinated Pt(II) complexes (−)-**3** and (−)-**4**, showing a calculated longer absorption wavelength than (−)-**1** and (−)-**2** with the same cyclometalated ligand (Appendix A), which accords with the ones observed in the experiment (Figure 3).

To gain an insightful understanding of emission, natural transition orbital (NTO) analysis has been accomplished to examine the S_0_→T_1_ excitation based on optimized T_1_ geometries (Figure 6 and Appendix A) [19,28]. The hole (H) orbital of (−)-**1** is mainly resident in the phenyl-pyridine or dipyridine moiety (**Ring A** and **Ring B**), and the one of (−)-**2** is spread over the whole N^N′*C ligand (amino N atom, **Ring A**, **Ring B**, and **Ring C**). For complexes (−)-**3** and (−)-**4**, the central Pt atom holds a considerable distribution (18.16% and 11.07%) of the H orbital in addition to the contribution of the N^C*N′ or N^N′*C parent. All complexes’ particle (P) orbitals are mainly distributed on the phenyl-pyridine or dipyridine moiety (**Ring A** and **Ring B**), with a predominant contribution of over 85%. Hence, the luminescence of (−)-**1** and (−)-**2** can be mainly assigned as a ligand-centered (LC) triplet state (^3^*π*,*π**) with minor CT character. In contrast, the phosphorescence of (−)-**3** and (−)-**4** originates from a mixture of ^3^*π*,*π** and ^3^MLCT (from Pt(II) atom to N^C*N′ or N^N′*C) emissive states.

### 2.5. Cytotoxicity

All the prepared 5,6-membered bis(metallacyclic) Pt(II) complexes, including chloride precursors, were evaluated for their cytotoxicity against human cancer cell lines K562, SGC-7901, BEL-7402, A549, and HeLa with cisplatin as the positive control [29,30]. From Table 3, the obtained half-inhibitory concentration (IC_50_) values range from 0.47 to 8.85 μM. N^N′*C-coordinated Pt(II) complex (−)-**2** with isocyanide displays significant cytotoxicity against the above five human cancer lines, which is better than that of cisplatin and comparable to those of pincer-type platinum(II) complexes Pt^II^(NCN′)Cl, Pt^II^(CNN′)Cl, (CNN′)Pt^II^(C≡CCH2R), (CNN′)Pt^II^(C≡NL)^+^ and [Pt(trpy)(NHC)]^2+^containing N-heterocyclic carbene (NHC) ligand [31,32,33,34,35,36]. All complexes with N^N′*C ligands provide a cytotoxic activity, so N^N′*C 5,6-membered bis(metallacycle) may be important for high cytotoxicity [32]. In addition, it can be inferred that the introduction of isocyanide and alkynyl may improve the cytotoxic effect by comparing the cytotoxicity results of Pt(N^N′*C)Cl, (−)-**2** and (−)-**4**. UV-vis spectroscopy has also checked the stability of the complexes obtained in DMSO and DMSO-PBS media [37]. The absorbance at 424 nm of the DMSO-PBS and DMSO solutions of (−)-**2** decreases by ca. 30% and 10% at 37 °C over 5 days, respectively (Appendix A). However, the low energy absorptions of the DMSO-PBS solutions of other complexes dramatically decrease by 50–80%.

### 2.6. Cell Imaging

Given the intriguing luminescence properties of 5,6-membered bis(metallacyclic) Pt(II) complexes, we tentatively explore their applications in cell imaging. The cultured HeLa cells were inoculated in glass-bottom cell culture dishes. After overnight culture, the cells were stained with 10 μmol/L phosphorescent Pt complexes in a DMSO/DMEM mixture (5/95) for 15 min. After the removal of extracellular fluorescent dyes through washing with PBS buffer, the cell samples were imaged using confocal laser scanning fluorescence microscopy in blue, green, and red channels [38,39]. As shown in Figure 7, the chloride precursor Pt(N^C*N′)Cl can efficiently permeate cells, showing bright blue luminescence and bright green emission, and the image in the red channel is not satisfactory enough. Other luminescent Pt(II) complexes can also rapidly permeate cells. However, the image performances are inferior to Pt(N^C*N′)Cl. It can be found that the luminescent Pt(II) complexes mainly distribute in cell membranes and are hard to stain the intracellular substances [38,39,40]. Complexes (−)-**1**, (−)-**2**, and (−)-**3** show excellent photostability in the DMSO/DMEM mixture (5/95) and can retain 97% of the emission intensity after 40 min irradiation (Appendix A) [41]. However, nearly 10% of chloride precursors Pt(N^C*N′)Cl, Pt(N^N′*C)Cl, and complex (−)-**4** have decayed upon 40 min continuous excitation.

## 3. Experimental

### 3.1. General Methods

All reagents were purchased from commercial suppliers and used as received. High-resolution ESI (HR-ESI) mass spectrometry spectra were acquired on Thermo Scientific (Waltham, MA, USA) Q Exactive Mass, Thermo Scientific Q Exactive Focus, and Aglient (Santa Clara, CA, USA) 7250 & JEOL-JMS-T100LP AccuTOF (Tokyo, Japan) Spectrometer. The ^1^H and ^13^C NMR spectra were obtained on Bruker (Mannheim, Germany) DRX-400 spectrometer. Coupling constants are provided in hertz. UV-vis spectra were measured on a UV-3600 spectrophotometer. Photoluminescence (PL) spectra were measured by a Hitachi (Tokyo, Japan) F-4600 PL spectrophotometer (*λ*_ex_ = 420 nm). Emission quantum yields (*λ*_ex_ = 420 nm) and lifetimes were measured on a HORIBA JY (Kyoto, Japan) system. The circular dichroism (CD) spectra in CH_2_Cl_2_ solution were recorded on a Jasco (Tokyo, Japan) J-810 spectropolarimeter at a scan rate of 100 nm·min^−1^ and 1 nm resolution at room temperature (using 10 mm quartz cell for the concentration of 5 × 10^−5^ mol·L^−1^, bandwidth = 1 nm, response = 1 s, accumulations = 3). Images of cells were obtained using a Nikon (Tokyo, Japan) A1 confocal laser scanning microscope.

### 3.2. Synthetic Procedures

#### 3.2.1. Preparation of Chloride Precursors

N^C*N′- and N^N′*C-coordinated Pt(II) chloride precursors were prepared according to our reported literature [19].

#### 3.2.2. Preparation of (−)-**1**

An equivalent of 2,6-dimethylphenyl isocyanide (18.3 mg, 0.14 mmol) dissolved in dichloromethane was added dropwise into a vigorously stirred dichloromethane solution of Pt((−)-N^C*N′)Cl (100 mg, 0.14 mmol) pre-covered by an aqueous solution of excess AgOTf. After stirring at RT for 1 h, the dichloromethane solution was separated, and the aqueous phase was extracted with dichloromethane (20 mL × 3). The organic phase was washed with brine water and then dried over anhydrous Na_2_SO_4_. After removing the solvent in vacuo, the final product was obtained as green-yellow powder. Yield: 80%. ^1^H NMR (400 MHz, CDCl_3_-d1): δ 9.21 [dd, *J*_1_ = 6.4 Hz, *J*_1_ = 1.2 Hz, 1H], 8.47 [s, 1H], 7.72 [t, *J* = 7.6 Hz, 3H], 7.62–7.67 [m, 2H], 7.45 [d, *J* = 7.2 Hz, 1H], 7.41 [t, *J* = 7.6 Hz, 1H], 7.33 [d, *J* = 7.2 Hz, 2H], 7.28 [t, *J* = 7.6 Hz, 2H], 7.13 [t, *J* = 7.6 Hz, 1H], 6.84 [td, *J*_1_ = 6.8 Hz, *J*_2_ = 1.2 Hz, 1H], 6.67 [d, *J* = 9.2 Hz, 1H], 6.40 [d, *J* = 8.0 Hz, 1H], 3.18 [s, 2H], 2.77–2.81 [m 2H], 2.59 [s, 6H], 2.40 [m, 1H], 1.45 [s, 3H], 1.30 [d, *J* = 9.6 Hz,1H], 0.73 [s, 3H]. ^13^C NMR (100 MHz, CDCl_3_-d1): δ 165.1, 157.9, 151.2, 150.5, 147.9, 147.8, 145.0, 142.0, 139.4, 138.5, 136.2, 132.3, 131.5, 130.4, 130.3, 129.4, 127.9, 123.0, 120.7, 120.3, 119.9, 118.2, 117.0, 45.1, 40.0, 39.9, 33.8, 32.0, 26.2, 22.0, 19.6. HRMS (ESI) (*m*/*z*): [M]^+^ calcd for C_38_H_35_N_4_Pt^+^, 742.2504; found, 742.2498. Anal. Calcd for C_39_H_35_F_3_N_4_O_3_PtS ((−)-**1**): C, 52.52; H, 3.96; N, 6.28; O, 5.38%. Found: C, 52.51; H, 3.95; N, 6.29; O, 5.38%.

#### 3.2.3. Preparation of (−)-**2**

The preparation procedure of (−)-**2** is the same as the one of (−)-**1**, we just changed the Pt(II) chloride precursor to Pt((−)-N^N′*C)Cl. The final product of (−)-**2** was also obtained as yellow powder. Yield: 90%. ^1^H NMR (400 MHz, CDCl_3_-d1): δ 8.73 [s, 1H], 8.49 [s, 1H], 8.44 [d, *J* = 7.6 Hz, 1H], 8.19 [dd, *J*_1_ = 7.6 Hz, *J*_2_ = 1.6 Hz, 1H], 8.00 [dd, *J*_1_ = 9.2 Hz, *J*_1_ = 7.6 Hz, 1H], 7.74 [t, *J* = 7.6 Hz, 2H], 7.66 [tt, *J*_1_ = 7.6 Hz, *J*_2_ = 1.2 Hz,1H], 7.43 [t, *J* = 7.6 Hz, 1H], 7.36 [d, *J* = 7.6 Hz, 2H], 7.31 [d, *J* = 7.6 Hz, 2H], 7.11 [td, *J*_1_ = 7.6 Hz, *J*_2_ = 1.6 Hz, 1H], 6.91 [d, *J* = 7.6 Hz, 2H], 6.68 [dd, *J*_1_ = 8.8 Hz, *J*_2_ = 1.2 Hz, 1H], 3.34 [dd, *J*_1_ = 7.6 Hz, *J*_2_ = 2.0 Hz, 2H], 2.88 [t, *J* = 5.6 Hz, 1H], 2.77–2.83 [m, 1H], 2.64 [s, 6H], 2.44 [m, 1H], 1.45 [s, 3H], 1.32 [d, *J* = 10.0 Hz,1H], 0.72 [s, 3H]. ^13^C NMR (100 MHz, CDCl_3_-d1): δ 154.9, 153.9, 151.4, 148.8, 147.8, 144.1, 142.8, 139.3, 138.6, 135.9, 131.9, 130.8, 130.2, 129.9, 129.0, 126.4, 124.6, 123.2, 119.8, 118.8, 117.2, 115.4, 45.1, 39.7, 39.4, 33.6, 31.4, 25.9, 21.7, 19.2. HRMS (ESI) (*m*/*z*): [M]^+^ calcd for C_38_H_35_N_4_Pt^+^, 742.2504; found, 742.2491. Anal. Calcd for C_39_H_35_F_3_N_4_O_3_PtS ((−)-**2**): C, 52.52; H, 3.96; N, 6.28; O, 5.38%. Found: C, 52.51; H, 3.96; N, 6.28; O, 5.37%.

#### 3.2.4. Preparation of (−)-**3**

A methanol solution of phenylacetylene (20.5 mg, 0.2 mmol) and sodium hydroxide (8 mg, 0.2 mmol) was stirred for 30 min at RT. Then, the chloride precursor Pt((−)-N^C*N′)Cl (97 mg, 0.15 mmol) was added to the above solution, reacting for a further 24 h. The solvent was removed in vacuo, and the solid was washed with methanol several times (20 mL × 3). The final product, a yellow powder, was obtained after recrystallization in a mixed methanol/chloromethane solution. Yield: 85%. ^1^H NMR (400 MHz, CD_2_Cl_2_-d2): δ 10.57 [dd, *J*_1_ = 6.4 Hz, *J*_2_ = 1.6 Hz, 1H], 9.67 [s, 1H], 7.68 [t, *J* = 7.6 Hz, 2H], 7.58 [t, *J* = 7.6 Hz, 1H], 7.53 [s, 1H], 7.49 [d, *J* = 7.6 Hz, 2H], 7.44 [tt, *J*_1_ = 7.6 Hz, *J*_2_ = 2.0 Hz, 1H], 7.33 [d, *J* = 7.2 Hz, 2H], 7.29 [m, 3H], 7.19 [tt, *J*_1_ = 7.6 Hz, *J*_2_ = 2.0 Hz, 1H], 6.95 [t, J = 7.6 Hz, 1H], 6.56 [td, *J*_1_ = 6.4 Hz, *J*_2_ = 1.2 Hz, 1H], 6.43 [d, *J* = 9.2 Hz, 1H], 6.20 [d, *J* = 8.0 Hz, 1H], 3.07 [d, *J* = 2.4 Hz, 2H], 2.89 [t, *J* = 5.6 Hz, 1H], 2.71–2.76 [m, 1H], 2.33 [m, 1H], 1.42 [s, 3H], 1.27 [d, *J* = 9.2 Hz, 1H], 0.73 [s, 3H]. ^13^C NMR (100 MHz, CD_2_Cl_2_-d2): δ 166.1, 159.2, 150.1, 149.9, 147.3, 147.1, 143.3, 143.2, 142.0, 139.9, 135.9, 135.5, 131.2, 130.4, 128.7, 128.6, 127.8, 124.9, 123.7, 118.3, 118.0, 117.7, 116.1, 114.4, 107.2, 44.5, 39.7, 39.2, 33.7, 31.6, 25.7, 21.3. HRMS (ESI) (*m*/*z*): [M]^+^ calcd for C_37_H_31_N_3_Pt^+^, 712.2166; found, 712.2105. Anal. Calcd for C_37_H_31_N_3_Pt ((−)-**3**): C, 62.35; H, 4.38; N, 5.90%. Found: C, 62.36; H, 4.39; N, 5.90%.

#### 3.2.5. Preparation of (−)-**4**

A 50 mL dry flask was charged with Pt((−)-N^N′*C)Cl (97 mg, 0.15 mmol), phenylacetylene (49.6 μL, 0.45 mmol), Et_3_N (1.40 mL), and CuI (2.1 mg) in chloromethane solution (15 mL). In the absence of light, the mixture was stirred at RT under argon for 24 h. The solvent was evaporated in vacuo, and the residue was purified by flash chromatography on the Al_2_O_3_ column with PE/EA (3/1, *v*/*v*) as eluent to yield an orange powder. Yield: 70%. ^1^H NMR (400 MHz, CDCl_3_-d1): δ 9.63 [s, 1H], 9.17 [dd, *J*_1_ = 6.4 Hz, *J*_2_ = 3.2 Hz, 1H], 7.85 [s, 1H], 7.56–7.66 [m, 5H], 7.55 [t, *J* = 7.6 Hz, 1H], 7.45 [d, *J* = 7.2 Hz, 1H], 7.35 [d, *J* = 7.2 Hz, 2H], 7.30 [t, *J* = 7.2 Hz, 2H], 7.18 [t, *J* = 7.2 Hz, 1H], 6.89 [dd, *J*_1_ = 7.0 Hz, *J*_2_ = 3.6 Hz, 2H], 6.68 [d, *J* = 9.2 Hz, 1H], 6.43 [dd, *J*_1_ = 6.4 Hz, *J*_2_ = 3.6 Hz, 1H], 3.08 [d, *J* = 10.0 Hz, 2H], 3.00 [t, *J* = 5.6 Hz, 1H], 2.75–2.80 [m, 1H], 2.37 [m, 1H], 1.45 [s, 3H], 1.28 [d, *J* = 9.6 Hz, 1H], 0.69 [s, 3H]. ^13^C NMR (100 MHz, CDCl_3_-d1): δ 155.1, 154.3, 148.9, 146.6, 146.0, 145.0, 144.2, 140.1, 134.5, 131.6, 131.1, 130.8, 129.4, 128.7, 127.9, 124.9, 123.7, 121.9, 121.3, 120.6, 118.2, 117.6, 113.1, 100.9, 44.9, 39.8, 39.3, 33.4, 31.5, 25.8, 21.5. HRMS (ESI) (*m*/*z*): [M]^+^ calcd for C_37_H_31_N_3_Pt^+^, 712.2166; found, 712.2060. Anal. Calcd for C_37_H_31_N_3_Pt ((−)-**4**): C, 62.35; H, 4.38; N, 5.90%. Found: C, 62.35; H, 4.39; N, 5.91%.

### 3.3. Single-Crystal X-ray Structure Determination

Single-crystal X-ray diffraction measurements were performed on a Bruker SMART APEX CCD and Rigaku (Tokyo, Japan) XtaLAB Synergy R. Intensities were collected with graphite monochromatized Mo Kα radiation (*λ* = 0.71073 Å) operating at 50 kV and 30 mA using ω/2θ scan mode. The data reduction was performed with the Bruker SAINT package [42]. Absorption corrections were performed using the SADABS program [43]. The structures were solved by direct methods and refined on F^2^ by full-matrix least-squares using SHELXL-2018/3 (Sheldrick, 2018) with anisotropic displacement parameters for all non-hydrogen atoms in the two structures. Hydrogen atoms bonded to carbon atoms were placed in calculated positions and refined as riding mode, with C−H = 0.93 Å (methane) or 0.96 Å (methyl) and Uiso(H) = 1.2 Ueq (C_methane_) or Uiso(H) = 1.5 Ueq (C_methyl_). All computations were carried out using the SHELXL-2018/3 program package [44]. CCDC numbers 2279411–2279412 contain the supplementary crystallographic data for this paper. These data can be obtained free of charge via https://www.ccdc.cam.ac.uk/structures/ (accessed on 5 July 2023) or by e-mailing data_request@ccdc.cam.ac.uk, or by contacting The Cambridge Crystallographic Data Centre, 12 Union Road, Cambridge CB2 1EZ, UK; fax: +44(0)1223-336033.

### 3.4. Calculation Methods

The crystal structures of (−)-**2** and (−)-**3** were used as starting geometries, and calculations were performed with the Gaussian 09 program [45]. Geometry optimizations of ground states were simulated with density functional theory (DFT) at the hybrid functional PBE1PBE-D3/LANL2DZ (Pt) and PBE1PBE-D3/6-31g(d,p) (H, C, N) levels using CH_2_Cl_2_ as solvent. The solvent effect is based on the polarizable continuum model (PCM). The optimized structures were used to calculate the lowest singlet electronic transition using the time-dependent density functional theory (TDDFT) method. The geometry of the first triplet state (T_1_) was optimized, and the analysis of the natural transition orbital (NTO) was carried out for the excitation of S_0_→T_1_ [28,46]. Mulliken population analysis (MPA) was utilized to obtain the electron density distribution of each atom in the specific molecular orbital of the Pt(II) complexes using the Multiwfn program [47].

### 3.5. Cytotoxicity

The cytotoxic activities of obtained 5,6-membered bis(metallacyclic) Pt(II) complexes were assessed against K562 (human leukemia cell line), SGC-7901 (human gastric carcinoma cells line), BEL-7402 (human hepatocellular carcinoma cell line), A549 (human non-small cell lung cancer cell line), and HeLa (human cervical cancer cell line) by MTT assay [29,30]. Briefly, the logarithmic phase cells were cultured in RPMI 1640 medium supplemented with 10% fetal bovine serum, 100 IU/mL penicillin, and 100 mg/mL streptomycin under conditions of 37 °C, 5% CO_2_, and 90% humidity. These human tumor cell lines with a density of 5 × 10^4^ unit/mL were seeded onto 96-well plates and then, after 24 h of incubation, treated with different concentrations of the sample dissolved in DMSO, respectively, while cisplatin was used as the positive control and DMSO was used as the negative control. After 72 h of incubation, MTT was dissolved at 5 mg/mL in PBS and used essentially as previously described. Finally, the inhibition rates were calculated using OD mean values measured by the MK3 Microtiter plate reader at 490 nm, and the IC_50_ value expressed as the mean standard deviation was determined using the Bliss method.

## 4. Conclusions

In summary, we have synthesized four groups of enantiomeric N^C*N′- and N^N′*C-coordinated Pt(II) complexes featuring a fused 5,6-membered bis(metallacycle). Their structures have been determined by NMR, HRMS, and single-crystal X-ray diffraction. Distorted square-planar coordination of the Pt(II) nucleus is observed for both isocyanide- and acetylide-containing derivatives. The solution of all complexes is emissive in the green-yellow or orange regions, and acetylide-coordinated Pt(II) complexes show a lower energy emission than those isocyanide-coordinated ones with the same cyclometalated ligand (Δ*λ* = ca. 20 nm in the solution). Furthermore, the difference between cyclometalated N^N′*C and N^C*N′ ligands induces a more significant effect (Δ*λ* = ca. 80 nm in the solution) on the emission. The influence trend is also observed for the solid-state emission. The emissive state of all complexes can be attributed to a ligand-centered (LC) triplet transition (^3^*π*,*π*) with some CT transitions. Platinum(II) complex (−)-**2** coordinated with isocyanide displays high cytotoxicity against the above five human cancer lines K562, SGC-7901, BEL-7402, A549, and HeLa. The existence of N^N′*C 5,6-membered bis(metallacycle) may be important for high cytotoxicity. All the complexes can efficiently permeate cells, mainly distributing in cell membranes and showing a clear cell outline. This research provides a reference for developing biologically active 5,6-membered bis(metallacyclic) Pt(II) complexes.

## Data Availability

Data will be available on request.

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
