# Peer review of "Synthesis and Biological Activities of Luminescent 5,6-Membered Bis(Metallacyclic) Platinum(II) Complexes"

_molecules, 2023, doi:10.3390/molecules28176369_

Round 1
Reviewer 1 Report
This manuscript by Jing et al. reports the synthesis and biological activities of luminescent five-six-membered metallacycle platinum(II) complexes. This study may contribute to the research fileds of luminescent materials and imaging probes, I suggest a minor revision.
1. For the structure characterization, MS spectra of the compounds should be provided in SI.
2. The PLQY of the compounds in solid state should be provided and compared with those in solution.
3. Excellent photostability is required for cell imaging, thus, the photostability of these compounds should be investigated and additional comments are needed.
4. Do the compounds possess photodynamic effect? Authors may consider this in their future study.
English language is fine.
Author Response
We thank the reviewer for her/his very careful reading and the resulting constructive comments. Here are the details of our response: 1. MS spectra have been added in SI. 2. The PLQY values of the compounds in solid state have been added. 3. The photostability of these compounds has been added. 4. Thank you for your constructive suggestion. The study aimed at the photodynamic effect is underway and will be reported in due course.Reviewer 2 Report
Congratulations nice paper. Spectra are lovely particularly with the inclusion of the solution and sample colors.
Probably good that the IC50s were not much of an improvement on cisplatin.
I only found one typographical error where there was not space between the temperature and the unit.
1. What is the main question addressed by the research?
Can the complexes be made and once made are thy cytotoxic and can they be monitored by the confocal microscope.
2. Do you consider the topic original or relevant in the field?
These are new complexes , so in that instance they are original.
Does it address a specific gap in the field?
They are fluorescent so may be useful in tracking their progress in the cells – even though they are about the same activity as cisplatin.
3. Are the conclusions consistent with the evidence and arguments presented and do they address the main question posed?
Yes
4. Are the references appropriate?
Yes
5. Please include any additional comments on the tables and figures.
They are appropriate and as noted in my response quite beautiful.
Author Response
We thank the reviewer for the positive comments on our manuscript. The typographical error has been revised.
Reviewer 3 Report
The manuscript submitted by X.-P. Zhang et al. is devoted to the synthesis and characterization of a range of new bis(metallacyclic) Pt(II) complexes, at least, some of which exhibited valuable luminescence and cytotoxic properties. In continuation of their own studies and related reports from other research groups, the authors present the results of investigations on the effect of structural modifications of the ligand framework on the properties of the resulting cyclometalated derivatives (extension of one of the fused metal-containing rings to obtain 5,6-memebered systems) bearing additional isocyanide or phenylacetylide ligands coordinated to the metal ions instead of the chloride one. This manuscript contains a sufficient amount of work with remarkable new results. It can be recommended for publication in the Journal after addressing the following issues.
1. The major point that raises certain questions is the stability of the complexes under consideration in DMSO and culture medium. This is quite important for any new type of potential metal-based cytotoxic agents. The stability of the complexes obtained in DMSO, DMSO-water, and DMSO-buffer media can be evaluated, for example, by UV-vis spectroscopy. The monitoring time must cover the whole incubation period used in the cytotoxicity assay. Note that the stability studies could shed some light on the possible reasons for different activities of the complexes explored.
2. The discussion of the structural characteristics of the metallacycles obtained must be supplemented by some comments on their NMR spectral features (preferably in comparison to those of the chloride precursors and 5,5-membered counterparts).
3. In Scheme 1, the substituents X1 and X2 must be written with superscript numbers since subscript figures are used for multiple substituents.
4. Please consider the following suggestions for a slight modification of the manuscript title: "Synthesis and biological activities of luminescent 5,6-membered bis(metallacyclic) platinum(II) complexes". Regardless of whether the authors will accept these changes or not (this is entirely at their own discretion), it should be noted that the figures designating the sizes of fused metallacycles should be separated by a comma rather than a hyphen symbol (without any spaces). This should be corrected throughout the manuscript text.
5. Actually the complexes obtained refer to a privileged class of pincer complexes. In recent years, the latter have attracted considerable interest as potential anticancer agents (see, for example, a recent review by S. Wu et al., Org. Biomol. Chem., 2021, 19, 5254 and references cited therein). In this context, I strongly recommend to include the appropriate references on the cytotoxic properties of the related pincer-type metallacycles and provide some comparison of the results obtained in this work with the activity of the related Pt(II) pincer complexes reported earlier.
6. To underscore the different nature of nitrogen donors an additional prime symbol can be used: N^C*N' and N^N'*C.
7. There are some unfortunate phrases and errors in the text. For example:
– both solution and solid à both solutions and solid samples;
– C−H bond Activation à C–H bond activation;
– The emissions of N^C*N 48 and N^N*C-coordinated Pt(II) phenylacetylene and isocyanide derivatives are highly emissive, with Φem exceeding 40% à The emissions of N^C*N 48 and N^N*C-coordinated Pt(II) phenylacetylene and isocyanide derivatives are highly intensive, with Φem exceeding 40%;
– and explored their applications in cytotoxicity and cell imaging à and explored their cytotoxic properties and application in cell imaging.
The revised version of the manuscript should be carefully spell-checked and grammar-checked before submission.
8. Checkcif reports for some of the complexes explored contain B-level alerts. These must be explained.
There are some unfortunate phrases and errors in the text. The revised version of the manuscript should be carefully spell-checked and grammar-checked before submission.
Author Response
We thank the reviewer for his/her thorough review and highly appreciate the positive comments on our results. Here are the details of our response:
- The stability of the complexes obtained in DMSO and DMSO-PBS media have been added.
- The discussion regarding NMR spectral features has been added.
- The substituents in Scheme 1 have been revised.
- The title and corresponding expressions have been revised.
- The references have been added and the comparison of ctotoxicity has been provided. Also, the cytotoxicity against human cancer cell lines BEL-7402, A549 and Hela has been added.
- The N symbol has been revised throughout the manuscript text.
- The manuscript has been checked carefully throughout the manuscript text and revised.
- I am sorry that I could not find B-level alerts in the checkcif reports. Please see the uploaded cif and checkcif files in supplementary materials.
Round 2
Reviewer 3 Report
In my opinion, the authors addressed all the required issues. Therefore, the revised version of the manuscript can be submitted for publication.